# Potential Global Distribution of *Paracoccus marginatus*, under Climate Change Conditions, Using MaxEnt

**DOI:** 10.3390/insects15020098

**Published:** 2024-02-01

**Authors:** Qing Zhao, Huiping Li, Chao Chen, Shiyu Fan, Jiufeng Wei, Bo Cai, Hufang Zhang

**Affiliations:** 1College of Plant Protection, Shanxi Agricultural University, Taigu 030801, China; zhaoqing86623@163.com (Q.Z.);; 2Technology Center of Taiyuan Customs, No. 1 Xieyuan Road, Jingyuan District, Taiyuan City 030021, China; 3Hainan Province Engineering Research Center for Quarantine, Prevention and Control of Exotic Pests, Haikou Customs District, Haikou 570311, China

**Keywords:** papaya mealybug, climate change, invasive pest, climate scenarios

## Abstract

**Simple Summary:**

The papaya mealybug, *Paracoccus marginatus*, is an invasive pest species found all over the world. It is native to Mexico and Central America. It can now be found in more than 50 countries and regions, seriously threatening the safety of the agricultural and forestry industries. In the current study, the potential global distribution regions of *P. marginatus* were predicted under current and future climatic conditions using MaxEnt. The results suggested that the highly suitable areas were mainly present in tropical and subtropical regions, including South America, southern North America, Central America, Central Africa, Australia, and South and Southeast Asia. Under four climate scenarios (SSP126, SSP245, SSP370, and SSP585) in the 2050s and 2070s, the total suitable areas will change very little. In addition, the results showed that the min temperature of coldest month (bio6) was the most important factor influencing the distribution of *P. marginatus*, accounting for 46.8% of all contributions. Overall, the current study can provide a reference framework for the future control and management of papaya mealybug and other invasive insect species.

**Abstract:**

The papaya mealybug, *Paracoccus marginatus*, is an invasive pest species found all over the world. It is native to Mexico and Central America, but is now present in more than 50 countries and regions, seriously threatening the economic viability of the agricultural and forestry industry. In the current study, the global potential distribution of *P. marginatus* was predicted under current and future climatic conditions using MaxEnt. The results of the model assessment indicated that the area under the curve of the receiver operating characteristic ( ROC-AUC) was 0.949, while the TSS value was 0.820. The results also showed that the three variables with the greatest impact on the model were min temperature of coldest month (bio6), precipitation of wettest month (bio13), and precipitation of coldest quarter (bio19), with corresponding contributions of 46.8%, 31.1%, and 13.1%, respectively. The results indicated that the highly suitable areas were mainly located in tropical and subtropical regions, including South America, southern North America, Central America, Central Africa, Australia, the Indian subcontinent, and Southeast Asia. Under four climate scenarios in the 2050s and 2070s, the area of suitability will change very little. Moreover, the results showed that the area of suitable areas in 2070s increased under all four climate scenarios compared to the current climate. In contrast, the area of suitable habitat increases from the current to the 2050s under the SSP370 and SSP585 climate scenarios. The current study could provide a reference framework for the future control and management of papaya mealybug and other invasive species.

## 1. Introduction

Invasive insects have a serious impact on agricultural productivity, forest resources, and human health due to their habits and locations that avoid detection but allow rapid reproduction and rapid dispersal [1,2]. Invasive species can have either active or passive modes of dispersal [3]. The global trade in plant materials could be an important passive mode for the spread of invasive species, such as scale insects [4,5,6]. Scale insects can be passively dispersed in conjunction with the movement of plant material. They have a cryptic appearance, behavior that makes them difficult to detect, the ability to fight and withstand pest control measures, and can successfully adapt to global change [7]. In Europe, this group includes 129 (ca. 30% of European scale species) invasive or potentially invasive species [8].

The papaya mealybug, *Paracoccus marginatus* Williams and Granara de Willink, 1992 (Hemiptera: Pseudococcidae), is a polyphagous pest that can feed on a wide variety of taxa, with host plant records for 158 genera in 51 families [9]. The adult females are 2.0–3.5 mm long, soft-bodied, elongate oval and slightly flattened; immature instars and adult insects are covered with mealy wax and adult females secrete wax filaments to form a protective ovisac for the yellow eggs. The adult males are short-lived, small insects with long antennae, three pairs of well-developed legs; membranous fore wings with reduced venation and hind wings reduced to hamulohalterae; a pair of long, white waxy filaments at the posterior of the abdomen; and no functional mouthparts. The female *P. marginatus* has three immature stages before molting to the lavaform adult stage. The male is likely to have two immature larval stages that feed, followed by non-feeding pre-pupal and pupal stages before it molts to a short-lived, winged adult [10]. In tropical conditions, the generations are not synchronized and there are several each year, possibly as many as 11 in favorable conditions [11]. This leads to curling of the leaves, yellow-colored leaves, the abscission of flowers and fruits, the production of a large amount of honeydew, and the honeydew causes sooty blotch, which eventually leads to death [12,13]. It has caused havoc in agricultural and horticultural crops, resulting in huge economic losses [14]. Since it was discovered in Coimbatore in 2008, *P. marginatus* has become an important pest for the majority of economic crops, such as papaya and Moraceae in India [15]. In September 2008, there were large infestations of papayas in Sri Lanka with papaya mealybug, with an average damage rate of 85.9% and significant associated losses for the local papaya industry [16].

The papaya mealybug is native to the Americas [17]. It was first reported in 1992 from the Neotropical region [18]. It became an invasive pest in the Caribbean Islands and USA (Florida) in 1994–2002; the West and Central Pacific islands in 2002–2006; South-East Asia and the Indo-Pacific islands in 2008–2010; West Africa in 2010–2016; East Africa from 2015; and Israel and Kenya in 2016 [19,20]. So far, *P. marginatus* has been recorded in more than 50 countries worldwide [21]. Its distribution will probably be limited by its cold tolerance, but as the planet warms, it is likely that *P. marginatus* will extend its range to countries further away from the equator [9,22].

The dispersal to uncolonized areas and subsequent outbreaks of *P. marginatus* are facilitated by the life history traits of the species. It has a short life span, high reproductive capacity, rapid rate of dispersal, and a broad range of host plants [15,23]. In order to limit the expansion of this invasive pest, suitable habitats need to be characterized and uninhabited areas that may be at risk should be identified. If the dispersal of the species can be predicted, strategies to restrict the introduction and spread of this invasive pest can be developed. Species distribution models (SDMs) provide one means by which identify the potential geographic distribution of many such invasive species [24,25,26]. Some of the SDMs that have been developed include Bioclim [27], genetic algorithm for rule-set prediction (GARP) [28], Climex [29], generalized linear models (GLMs) [30], artificial neural networks (ANNs), random forests (RFs) [31], and MaxEnt [32]. MaxEnt performs the best and is the most robust model when using small sample sizes and presence-only data to simulate potential species distributions [26]. It is also one of the most popular and widely used models, in comparison to other methods, due to the following advantages:(i)It uses presence-only occurrence data;(ii)It simultaneously uses continuous and categorical variables;(iii)It effectively controls the model fit through certain parameter settings;(iv)It relies on the present data, so that the sampling bias can be better dealt with [33].

This software was used in many other studies of invasive species, such as two scale insect species, *Dysmicoccus brevipes* (Cockerell, 1893) and *D. neobrevipes* Beardsley, 1959 [24], and two plant species, *Paeonia delavayi* Franch and *P. rockii* [34].

Some studies have shown that climate change is expected to alter the geographic distribution and abundance of invasive species by impacting the physiology and behavior of the species [35]. In our study, the current and future global potential distribution of *P. marginatus* was estimated based on the occurrence data. The major goals of the study were as follows:(1)To predict the trends in changes of suitable habitat areas under different climate change scenarios;(2)To identify the major climatic variables that restrict the potential distribution of papaya mealybug;(3)To provide a theoretical reference for policy makers to control and reduce invasive risks in future.

## 2. Materials and Methods

### 2.1. Occurrence Records Collection

The initial occurrence data were mainly collected from three sources accessed on 20 September 2023: (1) the Global Biodiversity Information Facility database (GBIF, https://www.gbif.org/ (accessed on 21 December 2023)); (2) the Center for Agriculture and Bioscience International (CABI, 2014, https://www.cabi.org/ (accessed on 21 December 2023)); (3) published literature [16,17,19,36,37,38,39,40,41,42,43]. The total number of occurrence records obtained using this approach amounted to 493 localities. All sites were converted into geocoordinates using Google Earth.

We processed the distribution data as follows: (1) first, we excluded distributions with no coordinates, zero coordinates, and duplicates; (2) second, we used the software CoordinateCleaner 3.0.1 [44] package to check whether the record points were around the capital, the center of the country, or whether they fell into the ocean, or were around museums that housed animals, and removed these problematic coordinate points; (3) third, to reduce sampling bias, the spThin 0.2.0 [45] package was used for spatial filtering so that each grid has only one distribution point using a spatial resolution of 2.5 arc minutes. We thereby obtained a total of 204 distribution records and projected them onto the world map using ArcGIS 10.7 (Figure 1).

### 2.2. Environmental Data

The set of 19 bioclimatic variables and elevation data were downloaded from the WorldClim database (https://www.worldclim.org/ (accessed on 21 December 2023)) at a spatial resolution of 2.5 arc minutes, which indicated the current temperature and precipitation conditions, including the minimum, maximum, and average values of temperature and precipitation values recorded from 1970 to 2000. Considering the effects of topographic factors, the altitude and slope variables were extracted from the elevation data in QGIS 3.12.2. In order to evaluate the impact of climate change on the species distribution, the authors also downloaded data for future climate predictions from the WorldClim database at a spatial resolution of 2.5 arc minutes. The future climatic data represented long-term mean climatic conditions for 2041–2060 (2050s), and 2061–2080 (2070s), respectively. Parameters for future climate data were chosen as follows: processed for the BCC-CSM2-MR global climate model, using four Shared Socio-economic Pathway (SSP) climate change scenarios (SSP126, SSP245, SSP370, SSP585).

### 2.3. Variable Selection

Variables are important for determining species niches in the environmental space. However, there is usually multicollinearity among multiple variables, which reduces the transferability of the model. This necessitates removal of some of the redundant and less important variables to minimize the effects of highly correlated variables. Pearson correlation coefficient (r) between variables was calculated using the R package corrplot 0.92 (Figure 2). Variables with |r|> 0.7 were considered highly correlated. Models that take into account information about species biology can increase model accuracy; we therefore factored it in when performing correlation analysis [46,47]. We also performed SDM analyses with all variables to test their percentage contributions using the jackknife test (Appendix A). The cold stress temperature and the limiting low moisture were the most sensitive parameters affecting the distribution of *P. marginatus* [13]. As bio19 was not highly correlated with all the remaining variables, we first selected it considering biological information and relevance. Second, we selected bio6 from inside the temperature factor, which was correlated with cold stress and highly contributing. Prolonged drought with scanty rainfall and fewer rainy days favored faster multiplication of pests [48]. Therefore, we selected bio17 from inside the precipitation factor that has a greater number of correlations with other variables and high contribution. The high contribution of bio13 was then selected from the remaining variables. Finally, we selected the highest contributing variable slope from the three topographic variables (elevation, slope, and aspect), which were not correlated with any of the variables. Finally, five variables (bio6, bio13, bio17, bio19, and slope) were selected for *P. marginatus*.

### 2.4. Modeling Procedure and Evaluation

MaxEnt is one of the most popular and widely used presence-only models, as it is simple to use and performs well. However, if the user only adopts the default output of model without considering the role of model optimization, the prediction results of the untuned model may have serious fitting bias. This might lead to incorrect assessment of the species niche. It will also mislead the formulation of related policies [49].

In general, the default settings of MaxEnt 3.4.4 might produce overfitted models [50]. The two main factors that affect the complexity of the MaxEnt model are the Feature Class (FC) and the Regular Multiplier (RM). The R package “ENMeval 2.0.4 [51]” was employed to choose the optimal combination of MaxEnt model parameters. The FC types include L = Linear, Q = Quadratic, P = Product, T = Threshold, and H = Hinge. Eight RM values range from 0.5 to 4, at increments of 0.5. A total of 64 parameter combinations of these RMs and eight FCs (L, LQ, LQP, QHP, LQH, LQHP, QHPT, and LQHPT) were used to calculate their Akaike information criterion coefficient (AICc) values. The parameter with the lowest delta AICc score (FC = QHPT, RM = 1) was considered optimal to run the model for *P. marginatus* (Figure 3).

The logical output was used for all analyses in MaxEnt 3.4.4 [52]. The suitable and unsuitable were subsequently determined by 10th percentile logistic training threshold, using 25% of the dataset for random testing, and 10-fold cross-validation was performed to prevent random errors. The jackknife test was used to assess effects of each environmental variable. The natural breaks (Jenks) method was used to reclassify the adaptive distribution into four classes: unsuitable, marginally suitable, moderately suitable, and highly suitable.

In recent publications, statistical evaluations of SDM predictions have generally been based on threshold-dependent and threshold-independent evaluations [53]. Common threshold-dependent evaluation indices include the Kappa coefficient and True Skill Statistics (TSS) [54], which is easily affected by the definition of threshold value and species distribution. The Receiver Operating Characteristics (ROC-AUC) [55] and the Boyce Index are threshold-independent indicators, which are not restricted by the threshold value or species distribution ratio. Therefore, they are widely applied in model evaluation as a type of robust assessment indicator. The AUC value ranges from 0 to 1, where a value < 0.5 can be interpreted as a random prediction; 0.5–0.7 as poor model performance; 0.7–0.9 as moderate model performance; and a value above 0.9 as a model with “good” discrimination abilities [56]. TSS values range from −1 to 1, with values closer to 1 indicating better model performance [54].

## 3. Results

### 3.1. Model Performance and Contributions of Variables

The results of the model evaluation were an AUC value of 0.949 and a TSS value of 0.820, both indicating good performance (Figure 4). The three variables that had the greatest impact on the model were found to be bio6, bio13, and bio19, with the corresponding contributions being 46.8%, 31.1%, and 13.1%, respectively (Table 1).

### 3.2. Potential Distribution under Current Climate

The logical output result generated based on the R package “maxnet” was expressed in terms of probability and ranged from 0 to 1. The result of the prediction was also divided into four levels, where 0–0.07 was considered to indicate an unsuitable habitat, 0.07–0.23 for a habitat of marginal suitability, 0.23–0.43 for a habitat of moderate suitability, and 0.43–1 for a highly suitable habitat (Figure 5). The climatic habitats of *P. marginatus* were distributed over all continents except Antarctica. Of the six continents, Europe had only a very small distribution of marginally suitable habitats, while on the other continents there was a wide range of climatic habitats. The highly suitable areas were identified as large areas in Asia (South-East Asia, South Asia), Africa (West, Central, and East Africa), Central America, South America, and only very sporadically in Oceania (Australia).

### 3.3. Potential Distribution under Climate Change

The total area of suitable areas increased in all scenarios except for the SSP126 and SSP245 climate scenarios, which decreased in the 2050s, with the largest increase in the 2070s under the SSP370 climate scenario (Figure 6). Based on the results obtained for the current distribution of *P. marginatus*, the areas of suitable habitat would change in accordance with future climate scenarios. These changes could be divided into three types, namely expansion, contraction, or no change. The overall spatial distribution of suitable areas for *P. marginatus* will change very little under future climate scenarios (Figure 7 and Figure 8). There is a tendency for the habitat to expand southwards in Argentina. In most cases, the suitable area in Mexico will expand northward to Texas in the United States and the suitable area in China will expand north-eastwards. In SSP585 this climate scenario, there is a larger block of expansion area in the Malaysian islands. The results of the centre-of-mass transfer suggest a future southward expansion of suitable areas for *P. marginatus* (Figure 9).

## 4. Discussion

### 4.1. Influence of Predictor Variables

Temperature, precipitation, and topographic factors were considered when simulating the habitat suitability for papaya mealybug in the current study. Among the five environmental variables used in the model, min temperature of coldest month (bio6), and precipitation of wettest month (bio13) were the variables that contributed the most to the construction of the model and hence the distribution of the pest. Thus, precipitation and temperature are important environmental factors that affect the growth and development of papaya mealybug, either directly or indirectly through the health of host plants. Previous research results have shown that rainfall and relative humidity were negatively correlated with the incidence of *P. marginatus* [57,58]. Seasonal rainfall and relative humidity might be influencing the distribution of this pest directly via effects on the activity of crawlers. However, papaya (*Carica papaya* L.), one of the main host plants of the mealybug, has a known negative relationship with seasonal rainfall and relative humidity [58], so there may also be an indirect impact on the presence or population size of the mealybug.

### 4.2. Changes in the Distribution of Papaya Mealybug in the Future

*P. marginatus* is considered as a major pest with multiple crop and forestry-related host species. In order to identify the key areas to control the pest, this study used the MaxEnt model to predict the potentially suitable areas for this pest worldwide under different climate change scenarios. *P. marginatus* has so far been recorded in the tropics and subtropics, including eastern, south-eastern, and southern Asia, western, eastern, and south-eastern Africa, and Central America.

The model of the current climate conditions showed that the areas highly suitable for *P. marginatus* had a wide global distribution range, including large areas in South Asia, Southeast Asia, Australia, Africa, Madagascar, Central America, and South America. The most suitable areas for *P. marginatus* are in the tropics and subtropics. The highly suitable area mainly includes parts of southern China (Guangxi, Guangdong, Hainan, Taiwan), the Philippines, Indonesia, Thailand, Cambodia, Laos, Vietnam, India, and Sri Lanka in Asia; Northern Australia in Oceania; Madagascar, Mozambique, Tanzania, Uganda, Kenya, South Sudan, Ethiopia, Somalia, Democratic Republic of Congo, Rwanda, Angola, Congo, Gabon, Central Africa, Cameroon, Nigeria, Benin, Togo, Ghana, and Cote d’Ivoire in Africa; Mexico, Guatemala, Honduras, Nicaragua, Cuba, Haiti, Jamaica, Dominica, and Panama in North America; Colombia, Venezuela, Ecuador, Peru, Brazil, and Bolivia in South America. In addition, Laos, Madagascar, Uganda, Ethiopia, Somalia, Democratic Republic of Congo, Rwanda, Angola, Congo, Cameroon, Colombia, Venezuela, Ecuador, Peru, Brazil, Bolivia, Jamaica, and Australia are highly suitable areas with no current distribution and need to be given high priority to strengthen prevention and control measures. These results could provide a framework with which to identify future quarantine and preventive measures for papaya mealybug. Many researchers have earlier predicted suitable areas for *P. marginatus* in the current climate using different models at global and local scales. The results obtained using the MaxEnt model by Song at the global scale were very similar to our predicted habitat areas, but differed in the degree of habitability [59]. In addition, its global invasion risk was also assessed using the CLIMEX model in combination with host data, and the predicted areas in South America were very similar to our study, whereas the suitable areas in Malaysia and Africa were larger than our stud [13]. At the local scale, climate suitability was assessed using the MaxEnt model for Kenya, Mozambique, and China. This study predicts a wider zone of suitability in Kenya, a smaller one in China, and only minor differences in Mozambique [36,59,60,61].

Under future climate scenarios, the change in the size of the suitable area is very small (no more than 6%). The area of suitable areas in 2070s increases under all four climate scenarios compared to the current climate. In contrast, the area of suitable habitat increases in the 2050s under the SSP370 and SSP585 climate scenarios. Insects are extremely sensitive to climate change and are affected by global warming. As a result of global warming, insects tend to spread to high latitudes (in the direction of the poles) or high altitudes, and their distribution will change significantly [62]. Changes in the size of these suitable areas are also reflected spatially. Southeastern China, northern Brazil, northern Argentina, and Uruguay have expanding areas in all scenarios, whereas Africa’s suitable area contracts very sporadically to the south and north. Texas in the United States is predicted to be climatically suitable, except for in the 2050s in the SSP126 and SSP245 climate scenarios. In addition, we also find that the Malaysian islands have an expanded area of habitability in the 2050s and the 2070s under the SSP585 climate scenario, which is consistent with the Song and Fitch results [13,59]. Finch et al. predicted the potential distribution of *P. marginatus* around the world using the CLIMEX model with spatial data for irrigation and cropping patterns [13]. The results showed that the areas of suitable habitat were widely distributed in Asia, Oceania, Africa, North America, and South America. The study also highlighted the potential expansion of the distribution area into novel areas in Central and East Africa, along with further expansion into Central America and Asia. In addition, this model also indicated that there could be small areas of highly suitable habitat in Spain and Italy, which was inconsistent with the results presented in our study. The main differences between current our study and previous studies are as follows: (1) the climate variables chosen by the two models are inconsistent; (2) the species distribution sites used by the two models are different; (3) the mechanisms of the two models differ in that CLIMEX predicts biologically appropriate climatic regions in which species can tolerate regional climates, whereas MaxEnt discovers points that are characterized by similar bioclimates to the training data [33,63]; (4) different species may be subject to the same variables, but at different spatial scales. Species with larger territories may be more vulnerable to landscape changes. Moreover, only localized studies of a type of species may lead to inaccurate predictions [64].

### 4.3. The Limitation of the Current Model

The results of model assessment proved the robust power of the current model. However, there are still some important factors affecting the potential distribution of species that have not been considered. For example, papaya is the main host of *P. marginatus*, so its distribution range will inevitably affect the invasion trend of this pest. In addition, some biotic interactions, such as interspecific and intraspecific competition, might also influence the distribution range of the species. Some other factors also affect the spread of this pest, like land use, vegetation, and human-mediated transport [65]. In addition, to these external factors, the dispersal capacity of the species itself will also affect its distribution range, which would depend on their ability to overcome geographical barriers and adapt to the new environment quickly [66]. Therefore, these factors should be the subject of future studies and included as model parameters to improve the accuracy and reliability of models used to determine the habitat requirements of the species.

### 4.4. A Comprehensive Control Strategy for P. marginatus

The long-distance transmission of this pest is a result of human mediation (females cannot fly); trade in agricultural commodities and the movement of seedlings through leaves, flowers, fruits, and stem parts that carry all insect forms, including eggs, larvae, pupae, and adults, is very important for its spread. The host, climate, and trade conditions are suitable for the growth, reproduction, and spread of the mealybug, and the large number of greenhouses and greenhouse cultivation in China also provide suitable environmental conditions for the survival of the insect. (1) For the non-epidemic suitable areas (such as Madagascar, Uganda, Ethiopia, Somalia, Democratic Republic of Congo, Rwanda, Angola, Congo, Central Africa, and Cameroon in Africa; Colombia, Venezuela, Ecuador, Peru, Brazil, Bolivia, and Jamaica in South America; and Australia in Oceania), risk warnings, quarantine permits, port inspection, isolation, and quarantine treatment should be strengthened. (2) Vigilance should be exercised against areas of expansion in future climates, e.g., northern Brazil, northern Argentina, Texas, Uruguay, southeastern China, Australia, and Malaysia. (3) In countries where the disease has already occurred, quarantine, field control, and eradication should be strengthened to prevent its further spread. Traditional biological control methods are used. Three species of specialized parasitoid, *Anagyrus loecki* Noyes and Menezes, *Acerophagus papaya* Noyes, and *Pseudleptomatix Mexicana*, have been used as natural enemies in Florida, Guam, and the Pacific Islands [67,68]. The populations were effectively controlled. Therefore, it is recommended that the availability of these natural enemies in the suitable areas of papaya mealybug should be assessed to prevent the spread and outbreak of the pest.

## 5. Conclusions

In conclusion, the papaya mealybug *P. marginatus* has attracted extensive interest in recent years as an invasive pest species. In this study, the global potential distribution of *P. marginatus* was predicted under current and future climatic conditions using MaxEnt. The three variables with the greatest impact on the model were bio6, bio13, and bio19, with corresponding contributions of 46.8%, 31.1%, and 13.1%, respectively. The MaxEnt results indicated that the highly suitable areas were mainly located in tropical and subtropical regions, including South America, southern North America, Central America, Central Africa, Australia, the Indian subcontinent, and Southeast Asia. Under four climate scenarios in the 2050s and 2070s, the area of suitability will change very little. Moreover, the results showed the area of suitable areas in the 2070s increases under all four climate scenarios compared to the current climate. In contrast, the area of suitable habitat increases from the present to the 2050s under the SSP370 and SSP585 climate scenarios. The current study could provide a reference framework for future control and management of papaya mealybug and other invasive species.

## Figures and Tables

**Figure 1 insects-15-00098-f001:**
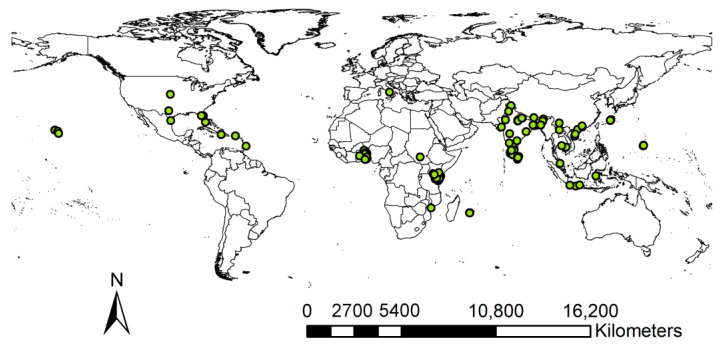
Worldwide geographic distribution records of *P. marginatus*.

**Figure 2 insects-15-00098-f002:**
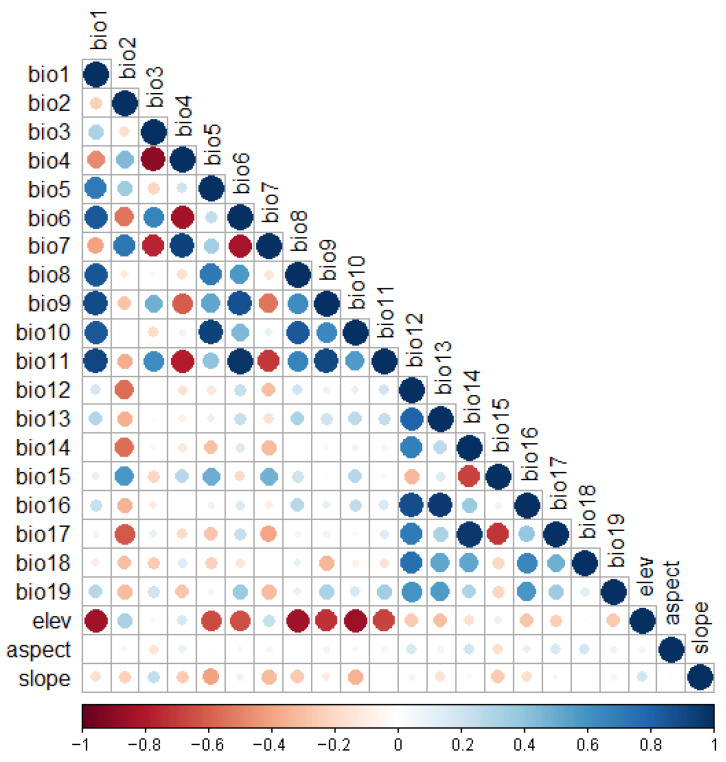
Pearson correlation analysis. Note: The horizontal scale represents the value of the correlation coefficient.

**Figure 3 insects-15-00098-f003:**
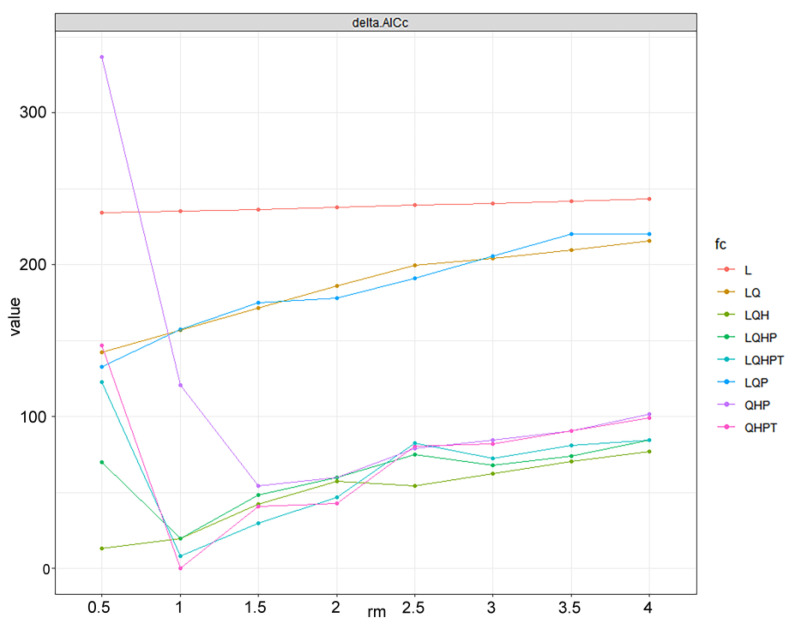
The AICc value of the parameter combination (FC, RM) calculated using ENMeval.

**Figure 4 insects-15-00098-f004:**
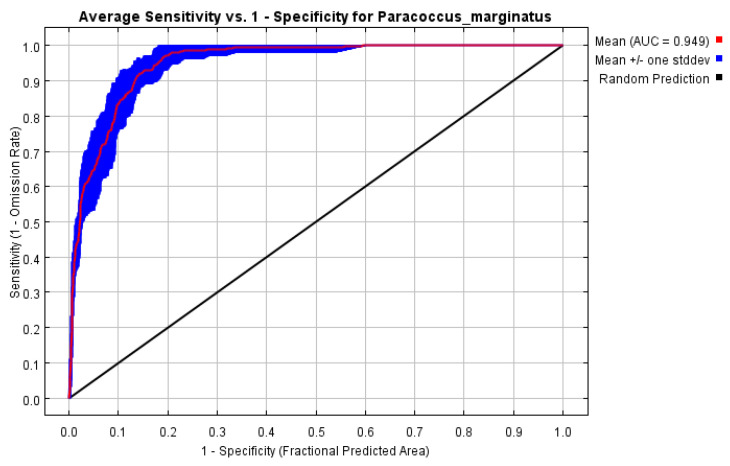
AUC curves for *P. marginatus* distribution models under current climate conditions.

**Figure 5 insects-15-00098-f005:**
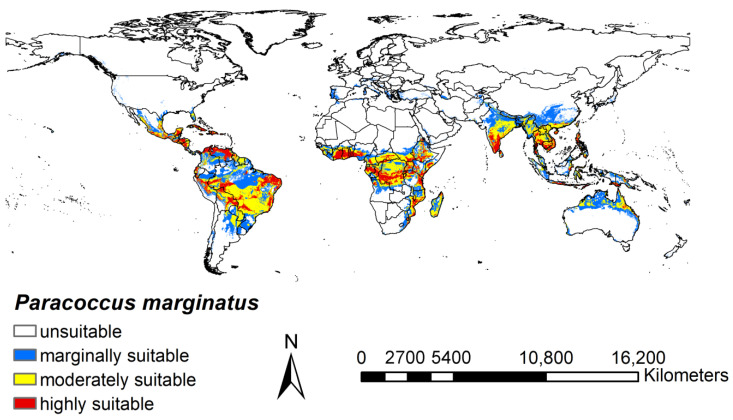
Potential distribution of *P. marginatus* under current climate conditions.

**Figure 6 insects-15-00098-f006:**
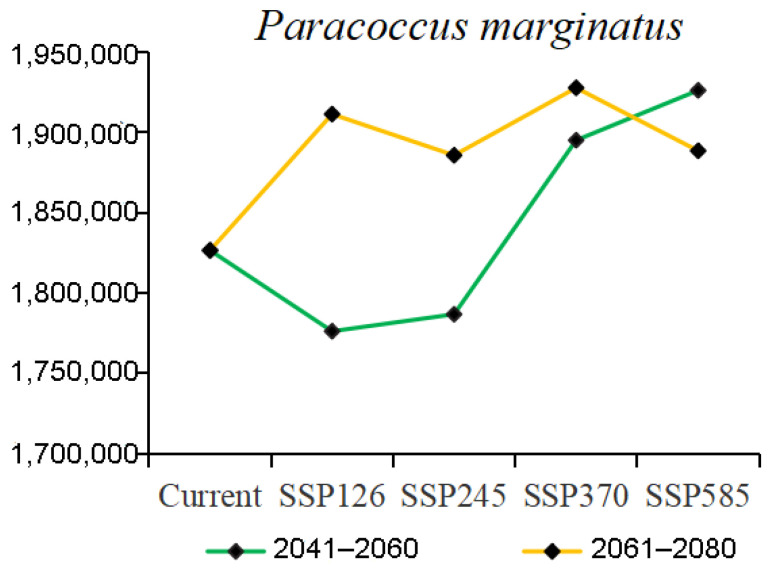
Changes in the total area with suitable climate for *P. marginatus* under future climate scenarios.

**Figure 7 insects-15-00098-f007:**
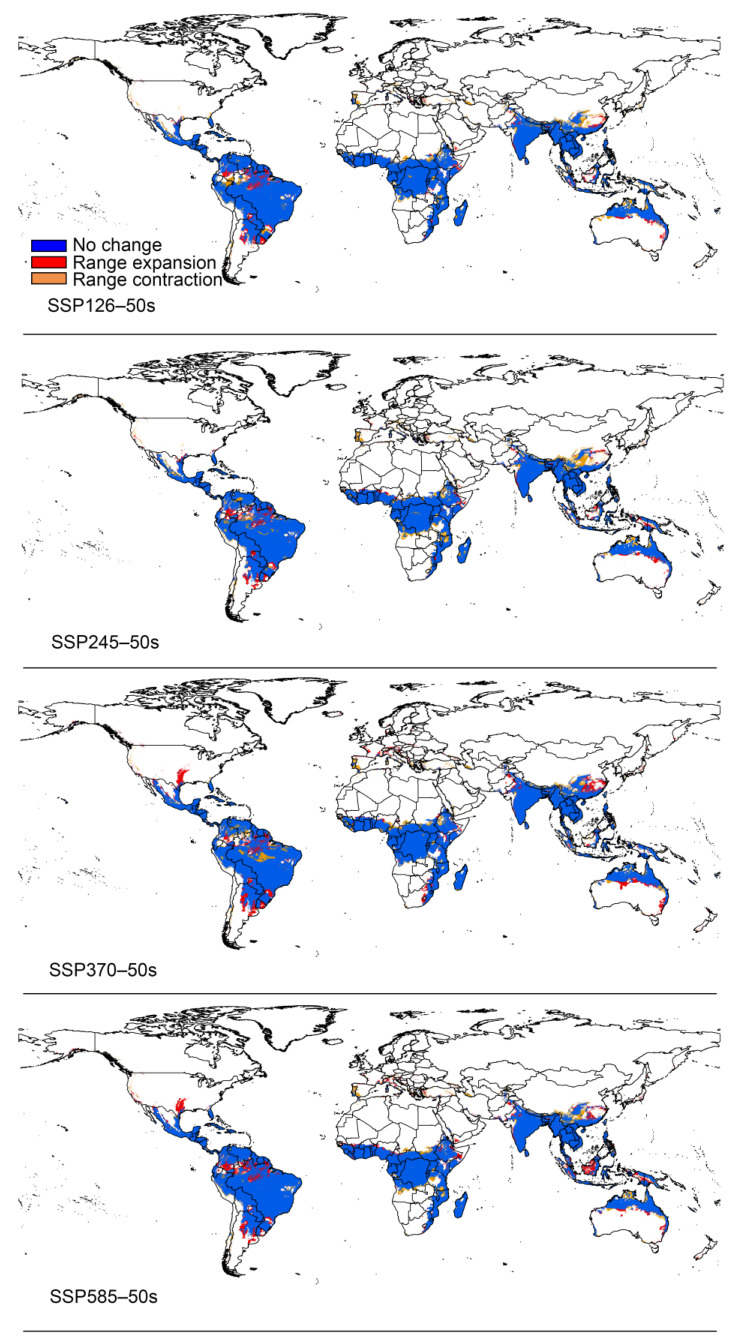
The potential distribution range of *P. marginatus* under different climate scenarios in the 2050s.

**Figure 8 insects-15-00098-f008:**
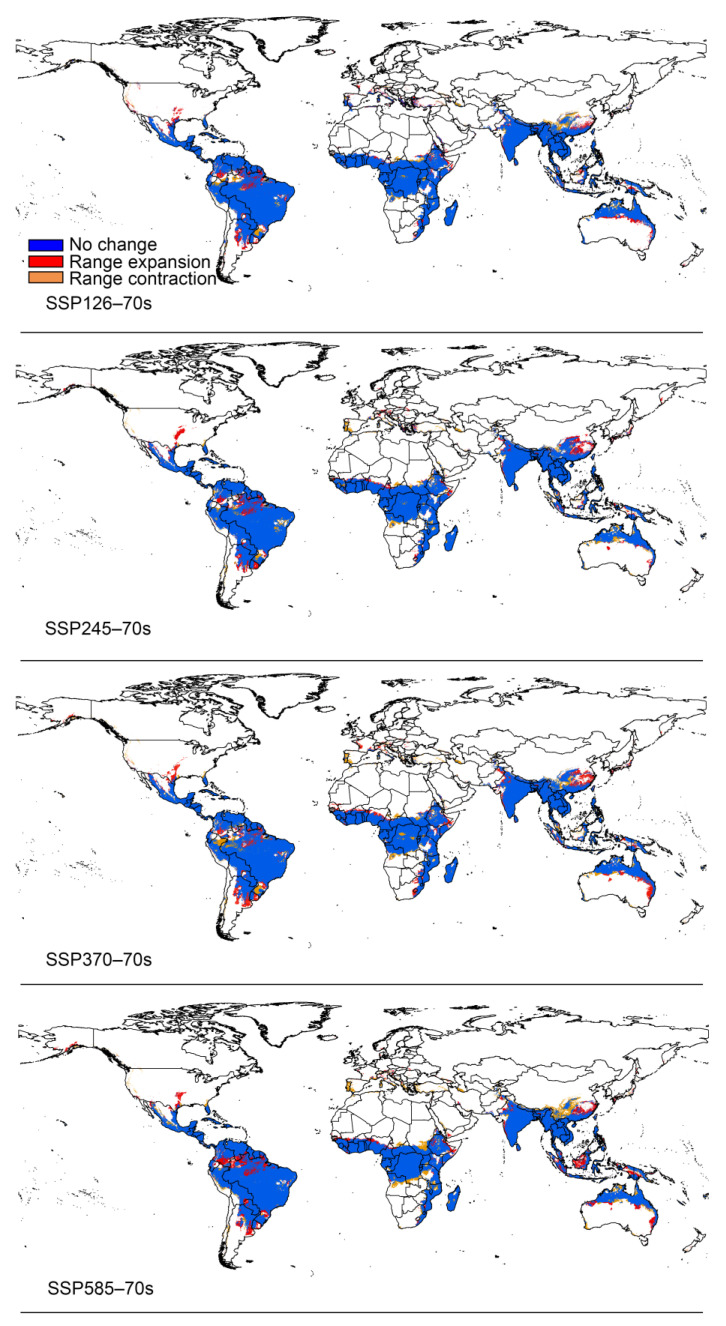
The potential distribution range of *P. marginatus* under different climate scenarios in the 2070s.

**Figure 9 insects-15-00098-f009:**
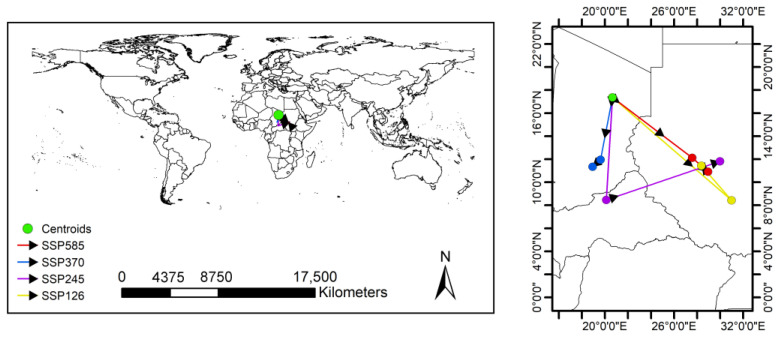
The centroids showing the geometric center of the distribution of *P. marginatus* under each scenario.

**Table 1 insects-15-00098-t001:** Relative contribution of each variable to the MaxEnt model using correlation analysis selection.

Variables	Percentage Contribution (%)
Min temperature of coldest month (Bio6)	46.8
Precipitation of wettest month (Bio13)	31.1
Precipitation of coldest quarter (Bio19)	13.1
Precipitation of driest quarter (Bio17)	7.3
Slope	1.6

## Data Availability

Available on demand.

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
