# Peer review of "Potential Global Distribution of Paracoccus marginatus, under Climate Change Conditions, Using MaxEnt"

_insects, 2024, doi:10.3390/insects15020098_

Round 1

Reviewer 1 Report (Previous Reviewer 3)

Comments and Suggestions for Authors

Authors made all corrections I ask them to make.

Author Response

Thanks for your guidance on our work, we have carefully checked the content of the article again and made some modifications.

Reviewer 2 Report (Previous Reviewer 2)

Comments and Suggestions for Authors

The manuscript has been greatly improved. The authors have completed information, as suggested in the review. However, some parts of text need correction.

Introduction

1.    The sentence: “on maturation she begins to secrete sticky, elastic, white wax filaments from the edges of her abdomen to form a  protective ovisac for her yellow eggs”

should be changed to the sentence: “immature instars and adult insects are covered with mealy wax and adult female secretes wax filaments to form a protective ovisac for the yellow eggs”.

2.     2 Please, change the sentence:

“The male is a short-lived, small insect with long, segmented antennae; six legs each bearing a single claw; one pair of simple wings coated with white wax powder; a pair of long, white waxy filaments at the posterior of the abdomen; and no mouthparts”

to:

“ The adult male is a short-lived, small insect with long antennae, three pairs of well-developed legs; fore wings membranous with reduced venation and the hind wings reduced to hamulohalterae; a pair of long, white waxy filaments at the posterior of the abdomen; and no functional mouthparts”.

The article should be cited:  Miller DR, Miller GL. 2002. Redescription of Paracoccus marginatus Williams and Granara de Willink (Hemiptera: Coccoidea: Pseudococcidae), including descriptions of the immature stages and adult male. Proceedings of the Entomological Society of Washington 104: 1-23.

3.     Most often Paracoccus marginatus has 11 generations per year.

The article should be cited: Seni, A., & Sahoo, A.K. 2015 Biology of Paracoccus marginatus Williams and Granara de Willink (Hemiptera : Pseudococcidae) on papaya, parthenium and brinjal plants Research on Crops 16 (4): 722-727

4     4. I do not understand the part of text: “can feed on multiple bituminous coal disease- locations of the host plant”.

Author Response

  1. The sentence: “on maturation she begins to secrete sticky, elastic, white wax filaments from the edges of her abdomen to form a protective ovisac for her yellow eggs” should be changed to the sentence: “immature instars and adult insects are covered with mealy wax and adult female secretes wax filaments to form a protective ovisac for the yellow eggs”.

Response: Thank you for your good advice, we corrected the description about female adult, and cited these related references.

  1. Please, change the sentence:

“The male is a short-lived, small insect with long, segmented antennae; six legs each bearing a single claw; one pair of simple wings coated with white wax powder; a pair of long, white waxy filaments at the posterior of the abdomen; and no mouthparts”

to:

“ The adult male is a short-lived, small insect with long antennae, three pairs of well-developed legs; fore wings membranous with reduced venation and the hind wings reduced to hamulohalterae; a pair of long, white waxy filaments at the posterior of the abdomen; and no functional mouthparts”.

Response: Thank you for your good advice, we corrected the description about male adult, and cited the related reference of Miller.

  1. Most often Paracoccus marginatus has 11 generations per year.

Response: Thank you for the reviewer's correction, we corrected it, and cited the related reference.

  1. I do not understand the part of text: “can feed on multiple bituminous coal disease- locations of the host plant”.

Response: Since this sentence is confused for readers, on the premise of not affecting the meaning and results of the article, we deleted it.

This manuscript is a resubmission of an earlier submission. The following is a list of the peer review reports and author responses from that submission.

Round 1

Reviewer 1 Report

Comments and Suggestions for Authors

The authors have attempted to model the potential distribution of the papaya mealybug using MaxEnt to look at its potential distribution now, but also for a future climate. Whilst the manuscript is generally well written (it does however need to be re-proofed as there were quite a few spelling grammatical errors), I think there are several fundamental flaws with the paper which make it currently unpublishable. These flaws relate to the data used in the model, and the modelling procedure for looking at future climates. 

Data used in the models:

I reviewed the distributional data on the PMB from GBIF, and once the necessary data cleaning had been carried out (removing those without coordinates, removing observations without coordinate uncertainty information/those with too large an uncertainty etc.), there were only 14 suitable records. I have also reviewed the data from the CABI website, and think that the authors are referring to national record data where the recorded coordinates are the central point of the country. This data is only supposed to represent that PMB is present within that country, and therefore it is not suitable for use with SDMs. The only other data I see on CABI is some archived data, but these are not enough to make up the 123 localities that the authors mention in the manuscript.

Further, he authors fall into the common trap of putting all possible variables into the model and removing those which are highly correlated or not important. This is the wrong method and can produce almost meaningless results. Instead, the authors should think about the life history of the PMB and choose the 4-5 most meaningful variables which can be included into the model. To me, adding in variable to do with soil characteristics is not useful, and included variables should focus on those which immediately affect the life history of the PMB - those relating to heat, cold, and rain.

Modelling procedure:

Whilst using climate change data for looking at how pest distributions might change in the future is a common practice, there are several flaws which arise in this paper. Namely, that the authors says that the PH of the topsoil and precipitation are both important factors affecting the pest distribution, but there is no allowance given that these will be affected by climate change - when looking at the affect of climate change, only a change in temperature is considered. Whilst I appreciate that it is impossible to obtain predictions of how precipitation and topsoil PH will look like in the face of climate change, stating that you are able to predict how CC will affect this pests distribution and then not taking into account how some of the most important variables will change, I’m afraid makes the results of the whole climate change aspect of this model almost meaningless.

My recommendation for the authors is to re-do the model making sure the distribution data has been throughly cleaned so only appropriate records are used, to think about 4-5 KEY environmental variables to include in the model (putting in every variable you can think of then removing unnecessary ones afterwards is bad modelling practice), and omit the climate change section of the model unless the environmental variables within your model can also be modelled with regard to climate change.

Further comments, are as such:

Line 58: needs a space between "locations" and "of". Furthermore, a space needs to be deleted before"flowers"

Line 63: I’ve never heard of “bituminous coal disease” - do the authors mean sooty coal?

Line 68-70: This sentence doesn’t make sense. Suggest changing to “In September 2008, there were large infestation of papayas in Sri Lanka with papaya mealybug, with an average damage rate of 85.9% and significant associated losses for the local papaya industry 

Line 73: delete full stop after 1990s.

Line 75: followed by where in 2002?

Line 97: Maxent does not de facto deal with sampling biases - these must be coded into the model. However, all the SDM techniques can be altered to account for potential sampling bias, so I’m not sure why the authors have insinuated that this is unique to MaxEnt

Line 142-144: The authors need to explain in much more detail how this was done.

Line 237: put a space between “model” and “without” 

Line 295-298: this needs a citation

Line 433-435: The authors need to explain better why they think that irrigation and crop distribution does not affect eh distribution of PMB - as a crop pest, I would think that the distribution of PMB is highly significantly affected by the distribution of crops. The authors even say later (line 448-449) that “For example, papaya is the main host of P. marginatus, so 448 its distribution range will inevitably affect the invasion trend of this pest”

Line437-438: The study by Finch et al. does account for spatial autocorrelation by filtering their data in the same way as the authors of this paper.

Reviewer 2 Report

Comments and Suggestions for Authors

The topic is interesting. Authors show important information regarding the potential global distribution of the invasive species. Methods and results are precisely described. The chapter "Discussion" includes all the necessary data on the distribution of Paracoccus marginatus.

I suggest to accept this paper with minor revision.

1. The chapter "Introduction" should be supplemented with short information on the number of generation per year, general morphology of adult female and male of Paracoccus marginatus and differences in life history between female and male.

Lines 99-101: Please correct author names of Paeonia rockii and remove date. It should be explained that there are two scale insect species and two plant species.

This reference should be cited in the chapter Introduction:

García Morales M, Denno BD, Miller DR, Miller GL, Ben-Dov Y, Hardy NB. 2016. ScaleNet: A literature-based model of scale insect biology and systematics. Database. doi: 10.1093/database/bav118. http://scalenet.info.

2. Authors should explain the differences between the number of occurrence sites used in their study and Finch's study. 

3. Other suggestions:

There are minor editorial mistakes in the text. Some species names are not  in italics in the chapters: Introduction (lines 99-101) and Discussion (line 361).

Lines 63-64: Is this correct: "death of the death"?

Figures are sometimes cited as Fig. (e.g. line 287) or Figure (e.g. line 298).

Some publication titles in References are written in capital letters (e.g. line 536-537).

The are no captions under the figures: S1, S2 and S3.

I suggest numbering the figures S1, S2 and S3 as figures 6, 7 and 8 respectively, since the current numeration is unclear in my opinion.

Drawings (Figure S3) should be enlarged.

There are many minor errors like missing space between words (e.g. 58). Authors should read their manuscript carefully and correct these mistakes.

Reviewer 3 Report

Comments and Suggestions for Authors

The manuscript is relevant and provides important information, especially for future scenarios of climate change in the global distribution of Paracoccus marginatus.As there is recent work published on almost the same subject (Finch et 2021) I suggest authors make it clear how their work differs from Finch et 2021) in the introduction.

I suggest that authors consult the articles below to enhance the discussion:

García Morales, M., Denno, B. D., Miller, D. R., Miller, G. L., Ben-Dov, Y., Hardy, N. B., 2019. ScaleNet: A literature-based model of scale insect biology and systematics. In: ScaleNet: A literature-based model of scale insect biology and systematics.http://scalenet.info/

Mendel Z, Watson GW, Protasov A, Spodek M, 2016. First record of the papaya mealybug, <i>Paracoccus marginatus</i> Williams & Granara de Willink (Hemiptera: Coccomorpha: Pseudococcidae), in the Western Palaearctic. Bulletin OEPP/EPPO Bulletin, 46(3):580-582. http://onlinelibrary.wiley.com/journal/10.1111/(ISSN)1365-2338

Watson, G. Paracoccus marginatus (papaya mealybug). CABI Compendium

https://doi.org/10.1079/cabicompendium.39201

More, suggestions are in the attached file
